# Heart Rate Variability Code: Does It Exist and Can We Hack It?

**DOI:** 10.3390/bioengineering10070822

**Published:** 2023-07-10

**Authors:** Martin Gerbert Frasch

**Affiliations:** Department of Obstetrics and Gynecology and Institute on Human Development and Disability, University of Washington School of Medicine, Seattle, WA 98195, USA; mfrasch@uw.edu

**Keywords:** HRV, brain–body communication, heart rate, phase space, health monitoring

## Abstract

A code is generally defined as a system of signals or symbols for communication. Experimental evidence is synthesized for the presence and utility of such communication in heart rate variability (HRV) with particular attention to fetal HRV: HRV contains signatures of information flow between the organs and of response to physiological or pathophysiological stimuli as signatures of states (or syndromes). HRV exhibits features of time structure, phase space structure, specificity with respect to (organ) target and pathophysiological syndromes, and universality with respect to species independence. Together, these features form a spatiotemporal structure, a phase space, that can be conceived of as a manifold of a yet-to-be-fully understood dynamic complexity. The objective of this article is to synthesize physiological evidence supporting the existence of HRV code: hereby, the process-specific subsets of HRV measures indirectly map the phase space traversal reflecting the specific information contained in the code required for the body to regulate the physiological responses to those processes. The following physiological examples of HRV code are reviewed, which are reflected in specific changes to HRV properties across the signal–analytical domains and across physiological states and conditions: the fetal systemic inflammatory response, organ-specific inflammatory responses (brain and gut), chronic hypoxia and intrinsic (heart) HRV (iHRV), allostatic load (physiological stress due to surgery), and vagotomy (bilateral cervical denervation). Future studies are proposed to test these observations in more depth, and the author refers the interested reader to the referenced publications for a detailed study of the HRV measures involved. While being exemplified mostly in the studies of fetal HRV, the presented framework promises more specific fetal, postnatal, and adult HRV biomarkers of health and disease, which can be obtained non-invasively and continuously.

## 1. Introduction

Brain–body communication is accomplished via multiple routes, e.g., neural, immune, metabolic, and endocrine, and is thought to be reflected in the mathematical properties of heart rate variability (HRV). Physiologic complexity is fundamental to HRV properties. As such, HRV analysis requires comprehensive assessment via complementary mathematical estimates from different signal–analytical domains [1,2]. What “comprehensive” means, in this context, turns out to be not obvious at all. Can a biomathematical concept of HRV as a code help us answer this question and advance our understanding of physiologic complexity in health and disease? The following is an attempt to advance this thought without any pretense of providing a comprehensive literature review of HRV studies. This has been carried out repeatedly already and is not the focus of the present work.

For example, psychophysiological models of HRV have been proposed and used intensely in research [3,4]. Many mathematical features of HRV have been reported [5], and for some mathematical HRV properties, a connection to physiological systems has been assumed, e.g., for RMSSD, modulations of HRV by the vagus nerve activity. Over the course of the past 40 or so years, a set of relatively distinct signal–analytical domains of HRV has been developed and deployed in various medical informatics settings [5,6,7].

Despite this progress in some fields, it remains striking that an overarching cohesive set of concepts has been lacking that addresses the question: How can we understand comprehensively and systematically the rich informational structure of HRV as a process-specific reflection of brain–body communication? Here, this issue is addressed by introducing and defining specifically the notion of HRV code based on a perspective from complex signals bioinformatics and rooted in a large body of experimental evidence [8,9]. A key component of a systematic approach to this question is to require the application of the same HRV analysis method across multiple preclinical and clinical studies to characterize multiple types of brain–body communication [5,7,10,11]. For this reason, most evidence is presented from the studies of near-term fetal and postnatal development. That body of work is used as the basis for the argument. However, the concept also generalizes observations in postnatal pediatric and adult HRV because the near-term fetus exhibits a rather mature autonomic nervous system (ANS) and HRV [12,13]. The approach is summarized in Figure 1.

A code is generally defined as a system of signals or symbols for communication [14]. To present the paradigm of HRV code in a systematic manner, this article is divided into sections on the experimental physiological evidence for the presence and utility of such code-like communication in HRV and its significance for future research and biomarker applications.

**Figure 1 bioengineering-10-00822-f001:**
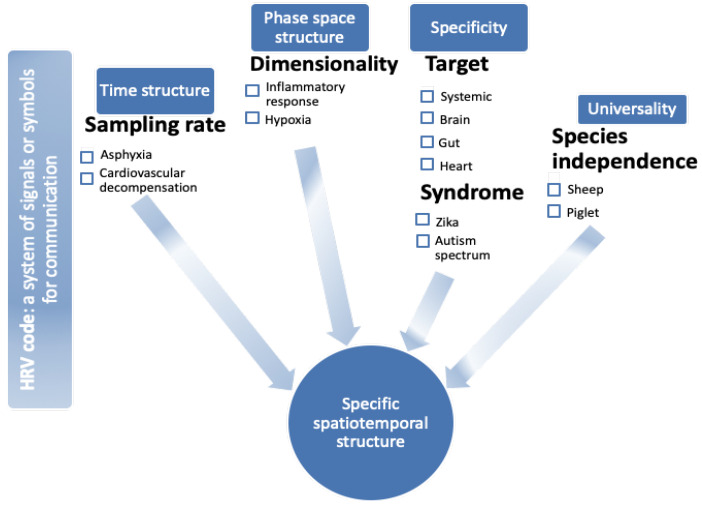
Heart rate variability (HRV) code: a visual abstract. HRV exhibits features of time structure [15,16,17,18,19,20,21], phase space structure ([7,8,22,23,24,25,26], specificity with respect to (organ) target [27,28,29], and pathophysiological syndromes [25,30,31], as well as species independence, i.e., universality with respect to species [32,33,34]. Together, these properties present the necessary features of an HRV code reflecting brain–body communication in the fluctuations of beat-to-beat intervals. Under each feature, (patho)physiological examples are provided, and respective references of experimental studies are provided and discussed in the article.

Briefly, HRV contains signatures of information flow between the organs and of response to physiological or pathophysiological stimuli as signatures of states (or syndromes). HRV exhibits features of time structure, phase space structure, specificity with respect to (organ) target, and pathophysiological syndromes, as well as universality with respect to species independence (Figure 1). Together, these features form a spatiotemporal structure, a phase space, that can be conceived of mathematically as a manifold of a yet-to-be fully understood dynamic complexity.

The focus of this review is to build upon the physiological experimental evidence for the notion of the existence of such mathematically defined HRV code and to illustrate its significance. Such a notion implies that the process-specific subsets of HRV measures indirectly reflect the phase space traversal reflecting the specific information contained in the code required for the body to regulate the physiological responses to those processes. While this work focuses on the abstraction of the concept, the interested reader is referred to the cited publications for a detailed look into the HRV measures involved and their mathematical properties.

This biomathematical view on HRV is novel and physiologically, translationally, and clinically relevant. After the presentation of the concept using an evidence/significance structure for each subthesis, this conjecture is synthesized in the conclusion section suggesting general directions for future research.

## 2. Time Structure: Sampling Rate

### 2.1. Evidence

HRV derived from ultrasound-based or electrocardiogram (ECG)-based sensors represents the key source of insights into prenatal, fetal, and postnatal wellbeing, development, and early detection of physiological abnormalities. There is no universally used heartbeat interval sampling method for the derivation of HRV data. In this section, it is shown that this status quo has consequences for our estimates of HRV properties, especially those relying on short-term time scales of brain–body communication reflected in HRV. An interested reader is referred to the following accompanying article [35].

In studies on chronically instrumented fetal sheep modeling human labor [15] and in human studies with HRV recordings during labor [16], we found that the sampling rate of the signal used to construct the beat-to-beat variability time series, i.e., the underlying electrocardiogram (ECG, 1000 Hz) or CTG (cardiotocography, 4 Hz) signals that yield HRV, impacts the precision of predicting the onset of acidemia, a fetal condition unique to labor and that is sometimes associated with unfavorable outcomes. Similarly, the early detection of fetal cardiovascular decompensation, a sentinel of incipient brain injury due to reduced cerebral blood flow, is impeded by the lower sampling rate of the underlying fetal ECG signal [17,18]. As putative underlying mechanisms, we propose reduced precision in capturing the coding of vagally mediated signaling in response to acidemia or the Bezold–Jarisch-like reflex, a vagally mediated adaptive cardiac depressor reflex preserving injured myocardium [19,36,37].

For any data capture technique following the Nyquist–Shannon sampling theorem, if the information is indeed encoded in the HRV signal, it must be sampled correctly for true representation to contain complete predictive information [38].

In several studies using animal models and human cohorts, we made observations validating this notion. Studies in pregnant sheep demonstrate the potential to detect fetal acidemia and predict cardiovascular decompensation during labor using high-precision ECG recording of the fetal heart rate (FHR) [15,17,19]. A novel bioinformatics approach to fetal HRV (fHRV) derived from maternal abdominal ECG during labor predicted well acid-base status at birth [16]. Conversely, HRV measures engineered to capture longer-term time scales of interbeat fluctuations recover the predictive information for early detection of fetal cardiovascular decompensation even when a 4 Hz sampling rate is used [20,21].

Fundamentally, the temporal coding in HRV is seen as an emerging consequence of the oscillations of multiple nonlinearly coupled oscillators, together contributing to the HRV’s dynamic properties in phase space, including time (Figure 2).

### 2.2. Significance

The evidence for temporal coding in HRV puts constraints on the potential for predictive utility of the low-sampling rate HRV tracking technologies such as photoplethysmography (PPG)-based wearables or ultrasound-derived fetal HRV monitors.

The impact of the duration and sampling rate of ECG on the quality and physiological representativeness of the ECG-derived HRV measures has been documented in part in the studies on adult subjects (cf. [40]) but has remained largely neglected in the fetal HRV literature, in part due to the sheer inaccessibility of high-quality fetal ECG technology in the clinical or research settings. Moreover, the sufficiency of a certain sampling rate below 1000 Hz, the gold standard, is usually inferred based on the HRV measures computed in the comparisons, which leads to intrinsically inconclusive proofs: one reports on the subset of HRV properties one has chosen to measure. With this in mind, the notion of time structure in HRV may appear not completely novel from the perspective of “adult” HRV studies, but it does represent the cutting edge in fetal monitoring.

Overall, within the integrative framework of the HRV code (Figure 1), the property of time structure exemplified but not limited to the effects of the sampling rate represents an intrinsic model component. Any reduction of the sampling rate below 1000 Hz must be seen as a compromise in the ability to capture completely the complex nonlinear oscillations contained in HRV. This has direct implications for the generalization of conclusions drawn from the respective studies.

### 2.3. Future Studies

The intrinsic limitations of the different sensor technologies must be accounted for during hardware and software stages of algorithm development with regard to their ability to adequately capture salient physiological information as well as the risk of capturing physiological artifacts at lower sampling rates [15,17,35,41].

In “adult” HRV studies, the advent of ubiquitous wearables as consumer devices and in decentralized clinical trials makes the aspect of the HRV time structure highly relevant because, so far, the systematic studies are lacking on the impact of the sampling rates in wearables versus conventional Holter-like ECG devices for specific health monitoring purposes.

The advantages of PPG-based wearables are the ease of use and versatility, as they typically double as smartwatches and have ubiquitous use. Hundreds of millions of PPG-based health trackers are already in use worldwide, and some attempts have been made, especially during the recent pandemic, to take advantage of the big data to extract some predictive information [42,43,44].

The key disadvantage of PPG-based wearables, and all wearables to date that are PPG-based, is their low sampling rate of physiological time series, such as beat-to-beat information and heart rate. The sampling rate differs by device, sometimes using a relatively higher, yet still physiologically insufficient, frequency internally than what is made available for analysis. Most importantly, the low sampling rate of heart rate impacts the resolution of heart rate fluctuations, as discussed above [35,40]. The ECG capability in such devices is either absent or very limited to user-initiated 30 s recordings. That is not intended and not sufficient for HRV analysis. Combined with varying internal data processing pipelines, device to device, it is challenging to date to perform any high-level HRV analyses.

Interestingly, very few options exist so far for consumer-grade ECG devices with a 1000 Hz sampling rate, the clinical gold standard. Some solutions do exist in sports, space, and military areas. Since this is not meant to be a review of commercial solutions available, the interested reader is left to perform a more specific search.

## 3. Phase Space Structure: Dimensionality of HRV

Here, the notion of dimensionality is used as it pertains to characterizing a system’s phase space [45]. The evidence is reviewed that HRV shows a complex, asymmetric phase space structure, i.e., including time dimension. Assuming that HRV activity represents an integrated set of properties of the physiological system, the information in the HRV is organized such that, with enough observations, HRV properties across the signal-analytical domains capture the entire phase space the system can traverse (Figure 3). Consequently, if the HRV code is adequately resolved, a set of deterministic equations and statistical or machine-learning models can predict the behavior of the underlying system.

An important and, due to its pleiotropic effects, enigmatic contributor to HRV properties is vagus nerve activity. The neuroanatomical distribution of the vagus nerve goes well beyond the well-known respiratory and cardiovascular controls as well as the more recently discovered metabolic and immunological control. The vagus nerve connects the brain with the body’s organs, in particular, the thymus, paraganglia (distributed sensor system in thorax and abdomen), liver, gastrointestinal tract, uterus, pancreatic islets, chemoreceptors sensing nutrients and related compounds (glucose, amino acids, fatty acids, and neuropeptides), mechanosensors (touch, tension, and serosal), temperature sensors, osmosensors, and nociceptors [27]. In contrast to this detailed understanding of vagal neuroanatomy, we are only beginning to understand the functional implications of this highly distributed system, notably in the regulation of nutrient signaling [47,48], glucosensing, and inflammation [22,23,24,25]. This is reviewed in the following section as it pertains to the notion of HRV code.

### 3.1. Evidence

There is evidence that physiological predictions are possible under the assumption of HRV phase space structure. Such predictions can shed light on the complex vagal nerve’s contributions to HRV.

Let us focus on the following physiological examples of HRV behavior:(1)The systemic inflammatory response [8,26];(2)Organ inflammatory response: brain and gut [49];(3)Chronic hypoxia and intrinsic (heart) HRV (iHRV) [50,51];(4)Allostatic load: physiological stress due to surgery [7];(5)Vagotomy (bilateral cervical denervation) [7,25].

The cholinergic anti-inflammatory pathway (CAP) signals via the vagus nerve’s afferent branch to surveil the body’s inflammatory milieu and relay this information to the brain [52,53]. Simply put, increased inflammation results in increased efferent vagal neural outflow, dampening inflammation. The afferent vagus nerve projects to nuclei in the brainstem, hypothalamus, amygdala, and insular and cingulate cortices, but the precise brain’s centers involved in processing afferent vagal information remain largely unknown. The indirect data reported so far suggest the existence of a neuroimmunological homunculus [25,43,54,55,56,57].

In contrast to the afferent and cerebral processing networks, the role of the efferent branch of the vagus nerve in the fine-grained control of inflammatory milieu under physiological and pathophysiological conditions has been studied more extensively. Underscoring its homeokinetic role, the research on the efferent branch yielded detailed insights into the anti-inflammatory, but not immunosuppressive, effects of the vagus nerve signaling on the spleen’s macrophages [53,58]. In parallel, evidence in animal models and human studies has accrued that such neuroimmunological signaling is also reflected in the specific changes in HRV properties [8,26,28,29,32,49,59,60].

### 3.2. Significance

The changes in HRV traverse a not well-understood phase space that can be, at least in part, characterized by complementary HRV measures from different signal-analytical domains (cf. Figure 3 and Table S1 in [7]). Such putative regions of the phase space capture specifically systemic or organ-specific, gut or brain, inflammatory responses to lipopolysaccharide (LPS) because they reflect the underlying HRV code of brain–body communication and may reflect indirectly the vagus nerve signaling measured via vagus nerve electroneurogram (VENG) [8,26,49,61].

In addition to predicting inflammation, another subset of HRV measures characterizes the iHRV properties imprinted by chronic exposure to hypoxia [50]. A complex landscape of HRV measures characterizes the chronic effects of stress imposed on the body due to surgery, and that landscape shifts subtly but distinctly when the surgery is conducted with the initial bilateral cervical removal of the vagus nerve [7]. Notably, in this shift, the most commonly used HRV metric of vagal modulations, RMSSD, is not altered.

### 3.3. Future Studies

Systematic studies in various species with high semblance to human physiology, such as pigs, sheep, and non-human primates, as well as clinical studies, are needed to develop fundamental physiological and translational pathophysiological insights into HRV phase space structure.

Physiological processes of special interest are sleep and exercise, with sleep being the best source of high-quality data, especially when wearables or ambient biosensors are used [30].

Pathophysiological processes of particular interest, due to their generic nature in multiple clinical syndromes, are acute and chronic inflammation (bacterial or viral), sepsis, and hypoxia/ischemia. Due to the HRV’s potential to describe systems’ behavior at an integrative level, special attention should be paid to studies investigating multiple hits and the identification of memory signatures reflecting previous exposures [31,50,62,63].

Such studies would account for the sympathetic and parasympathetic, especially vagal modulations of HRV as well as the more general effects of modulatory influences via intrinsic HRV, ANS, hormonal, and mechanical oscillations and entrainments via interactions with other systems (Figure 2 and Figure 3). Well-defined HRV processing pipelines need to be deployed throughout such studies to ensure comparability of the findings and to identify fundamental phase space structures likely present across species (cf. Section 5). Figure 3 provides an example of such an approach. Examples of comprehensive HRV processing pipeline implementation can be found on PhysioNet or in this study [30].

It should be noted that the various types of devices used for acquiring HRV data influence the quality and reproducibility of the HRV metrics and machine learning models developed in such studies. This constraint is discussed in part above, in Section 2.3. Future studies should explore specifically which HRV metrics are preserved in their fidelity with regard to specific outcomes using certain hardware technologies, such as the various PPG, ECG, radar-based ambient, or ballistocardiography (BCG) contact-free heartbeat detection technologies [64,65,66,67,68,69,70]. It is likely that certain physiological and pathophysiological trends can be captured with sufficient fidelity using low sampling rate PPG-, BCG-, and radar-based technologies as they can take advantage of days, weeks, and months of available data, especially during night-time. A systematic approach to discovering such use cases and identifying the limitations is needed in future work.

## 4. Target and Syndrome Specificity

### 4.1. Evidence

As we discussed above, for organ-specific gut and brain inflammation, no HRV measure correlated to several markers of inflammation at the same time.

These organ- or target-specific observations in gut and brain inflammation suggest the possibility of identifying further target-specific HRV signatures. While the attempts to identify HRV code represent an indirect attempt to hack brain–body communication, the direct data obtained from VENG studies are also mounting to suggest the existence of a vagus code [22,71,72]. It is plausible to expect a reflection of such vagus code in HRV properties that vagus activity is known to modulate.

If we are to postulate code properties, target specificity must be complemented by another kind of unique encoding property: specificity to complex physiological or pathophysiological behavior or a syndrome. To that claim, let us mention once more the finding of iHRV as well as the recent findings of HRV changes in toddlers exposed to Zika during gestation and eight-year-old children with autism spectrum disorder (ASD), conduct disorder, and depression. This is discussed in the following paragraphs.

Sheep fetuses exposed to chronic hypoxia in vivo and their hearts mounted in Langendorff apparatus ex vivo showed changes in cardiac activity: the beat-to-beat variability of the isolated hearts from hypoxic fetuses demonstrated the existence of a signature of such exposure [50].

Interestingly, the mathematical properties of the HRV measures comprising this signature were also found in an independently conducted prospective study of human toddlers born without overt symptoms to mothers exposed to the Zika virus during pregnancy [31]. Zika virus infection can cause chronic hypoxia via its effects on placental function, so the iHRV signature of fetal hypoxia is relevant for the changes in HRV due to Zika [73]. Together, these findings suggest that chronic hypoxia alters the cardiac pacemaker cellular activity and synchronization, impacting the properties of the emerging beat-to-beat variability. Another possible impact is on the normal developmental program of the pacemaker cell genes such that the resulting beat-to-beat variability is imprinted by the exposure carrying de facto a hypoxia memory [50].

In the cohort of 68 children of eight years of age, machine learning models built from features of select HRV measures identified children from healthy, ASD, conduct disorder, and depression cohorts with the area under the receiver operating curve of more than 0.82 [33]. The HRV feature selection was based on the above-mentioned preclinical studies and derived from five signal-analytical domains computed from five-minute ECG recordings (Figure 4).

### 4.2. Significance

The fundamental role of the vagus nerve in ASD has been the subject of ongoing research, notably in the polyvagal theory [3,34,74]. The above-mentioned findings in 68 children highlight the potential of the HRV code approach to capture complex traits or syndromes.

It has been proposed that ASD represents a process defined by a set of developmental immunometabolic constraints that may explain much of its features [75]. HRV has been shown to exhibit dynamic patterns with metabolic challenges [76,77,78,79,80]. Does HRV reflect metabolic status and metabolic optimization, as do genomic adaptations? [81,82,83].

### 4.3. Future Studies

More studies are needed to connect specific VENG features with multi-dimensional HRV properties, as was attempted recently [7,61,84].

Further studies are needed to better understand the impact of various insults on the developmental and functional properties of individual and synchronized cardiac pacemaker cells yielding heartbeat automaticity and intrinsic variability. This will help develop a stronger causal understanding of how the various environmental exposures impact the HRV properties with regard to HRV code target specificity.

Future studies should systematically investigate the relationship between HRV code and immunometabolism, especially in subjects with ASD or neurodegeneration, as these neurological conditions show altered metabolic signalling.

Can this information now be used to predict trajectories toward these conditions at an earlier age so as to help commence more timely interventions? If the HRV code persists and has memory properties, this should be a reasonable expectation and tested in future studies.

## 5. Universality

### 5.1. Evidence

Another anticipated aspect of an HRV code should be its universality. The patterns of communication reflected in the HRV code should carry over between species, at least to the extent that they represent some phylogenetically preserved and functionally parsimonious solutions. That is, it seems likely that complex systems would converge onto similar, if not identical, solutions to a problem such as encoding multiorgan communication, especially when relying on a similarly implemented substrate, such as the vagus nerve. We tested this in a study on newborn piglets exposed to a sublethal dose of LPS [61,85,86]. We applied the same HRV inflammatory index known to track systemic inflammatory response to LPS in fetal sheep [8,26] to the newborn piglet’s HRV. We found that the HRV inflammatory index tracked the inflammatory response in this species correctly, following the temporal inflammatory profile of the measured cytokines (Figure 5). Moreover, the same mathematical signature of inflammation also tracked the inflammatory response directly on VENG. These findings encourage the notion of HRV code universality, at least in the application of tracking inflammation.

### 5.2. Significance

An important aspect of the study design when investigating the universality of HRV code is the known between-subject variability of HRV measures [87]. This can be compensated by a within-subject design with repetitive measurements creating within-subject temporal profiles coordinated between subjects in terms of diurnal data collection, as, for example, carried out in the above-mentioned fetal sheep and piglet studies or in the review by Laborde et al. [87].

### 5.3. Future Studies

Based on the suggestions for experimental design made in Section 3.3, further studies are required to test the observations of species universality more extensively across different species, exposures, and outcomes and relate them to the corresponding direct changes in VENG.

An important limitation currently preventing the full translation of preclinical studies into the clinical realm is the fact that most, if not all, preclinical studies use traditional high-quality ECG electrophysiological setups, while clinical studies, except some ICU datasets, are typically relying on much lower sampling rates of heart rate (and other physiological data) [88]. Human clinical data are also often inaccessible for third-party analyses, but notable developments exist that have increasingly overcome these issues, such as the PhysioNet resource [89,90,91]. During the species comparative studies of HRV metrics and their predictive potential, special attention will need to be paid to the exact pipelines used to record and process the data, from the biosensors used to the artifact removal and any other signal modifications preceding the HRV computations.

It is exciting to consider the role of HRV metrics, across species, as part of a larger deep phenotyping effort to understand the physiology across the scales of organization, from (epi)genomic and microbiomic variability all the way to the systems level physiological variability manifested in HRV, its phylogenetically preserved, and species-specific traits [35,92].

## 6. Conclusions

This focused review is meant to generate experimentally testable hypotheses. Some future areas of interest have been highlighted throughout the review.

Overall, we observed that HRV exhibits a rich spatiotemporal structure if captured at adequate temporal resolution. That is, HRV properties are phase-space- and, in particular, time-resolution-dependent. These insights inform practical considerations of data acquisition, e.g., in the choice of wearables or ambient biosensors and beat-to-beat sampling rate, and the subsequent analysis, such as the choice of HRV measures and the statistical or machine learning models to predict the outcome or classify subjects based on the outcome. Some applications of machine learning were mentioned both for prediction [17,19] and classification [33] problems using HRV data while accounting for these considerations.

The subsets of HRV measures relate to specific physiological responses and hence are akin to code with regard to target specificity. This has implications for HRV data interpretation and biomarker discovery using HRV technology.

A fascinating question needs to be answered in future studies: after more than 50 years of studying HRV behavior under various conditions, can we bring this knowledge base together with the emerging field of bioelectronic medicine and its many promising therapeutic approaches to create a closed-loop system where HRV or VENG code hacking is used to sense the behavior of the system of interest and the vagus nerve stimulation or other bioelectronic tools are used to modify its phase space dynamics?

Reproducing the above observations under different physiological conditions and in different species will help address the issue of HRV code robustness. Does the HRV code benefit from stochastic resonance [93]? Can this property be measured in HRV, in vivo, or in silico?

At last, while the above considerations are driven by the pragmatic pursuit of HRV code applications for predictive purposes of time series, there remains the big question of what HRV code really represents physiologically. We can take inspiration for conceptualizing HRV code from the brain’s default-mode network concept. [94]. This question has recently been pursued in several studies [95,96,97,98].

Does the HRV code exhibit spontaneous intrinsic dynamics that are altered by external stimuli? It is suggested that the answer to this can already be stated in the affirmative. Progress has been made in defining the cardiac origins of HRV, arising from stochastic fluctuations of cardiac pacemaker cells’ ion channel activity, their networked synchronization, and modulation of this synchronization dynamics by the milieu intérieur and external neural and humoral inputs [99]. Future studies will address the questions of whether we can formulate the exact gestalt of such intrinsic dynamics driven by homeostasis and inter-organ communication on the one hand and define the activation patterns within the HRV code in response to external stimuli on the other hand.

## Figures and Tables

**Figure 2 bioengineering-10-00822-f002:**
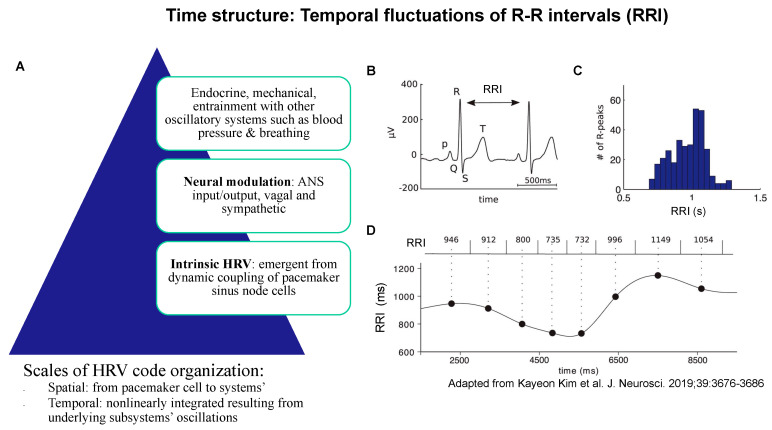
Time structure: temporal fluctuations of R-R intervals (RRI) of ECG arise from the spatiotemporal organization of physiological influences on pacemaker cell activity (**A**); together, they represent the property of the time structure of the HRV code. (**B**) An example ECG trace showing the QRS complex. RRI is defined as the time between two successive R peaks. (**C**) Histogram of RRI distribution in one patient. (**D**) The RRI time series (bottom) is constructed by assigning each RRI (top) to the central time point (large black dots) between two heartbeats, then interpolating the points, e.g., using a cubic spline function. Adapted from [39].

**Figure 3 bioengineering-10-00822-f003:**
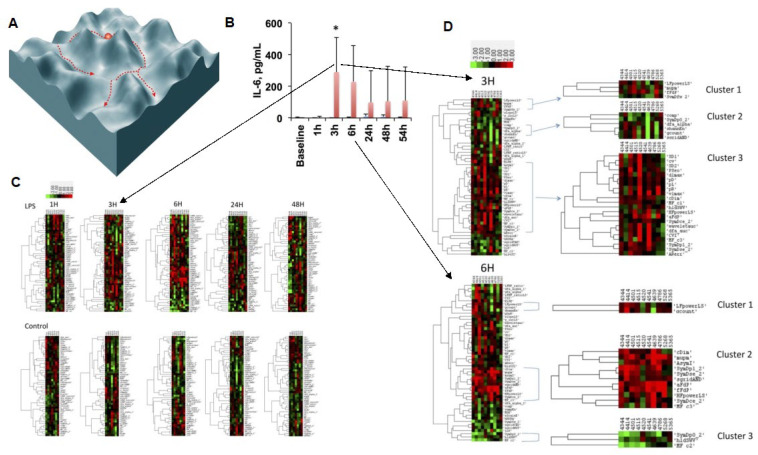
Phase space structure of HRV code: the traversal of phase space of the inflammatory space encoded by HRV is represented symbolically at first (**A**) and then exemplified by the findings of an HRV inflammatory index which tracks the temporal profile of the cytokine (IL-6) response to LPS exposure (**B**) over the period of 54 h (**C**) and zoomed in onto the temporal segment of most pronounced changes in the HRV metrics landscape defining the phase space (**D**). (**A**) Reproduced from Figure 2 in [46]; (**B**,**C**) reproduced with modifications from [8]. Details on HRV metrics can be found in [7,30]. For a detailed view of the HRV metrics on the Y axes of the heatmaps, please view the high-resolution version of this figure.

**Figure 4 bioengineering-10-00822-f004:**
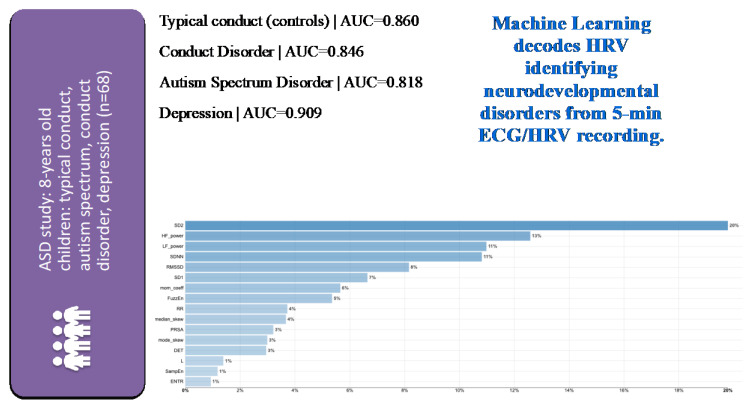
Identification of children with typical or specific neuropsychiatric disorders from 5 min ECG-derived HRV analysis using machine learning (ML) with a multi-class classification approach on HRV metrics selected a priori across signal-analytical domains based on their known role in reflecting inflammation or hypoxia [33]. Note the very good performance of ML models for each of the subcohorts as indicated by the area under the curve (AUC) metric. HRV metrics are ranked in the order of model importance. Details on HRV metrics can be found in [7,30]. The latter also contains a freely accessible Python codebase for further studies using comprehensive HRV pipelines across all signal–analytical domains. For a detailed view of the HRV metrics on the Y axes of the feature importance plot, please view the high-resolution version of this figure.

**Figure 5 bioengineering-10-00822-f005:**
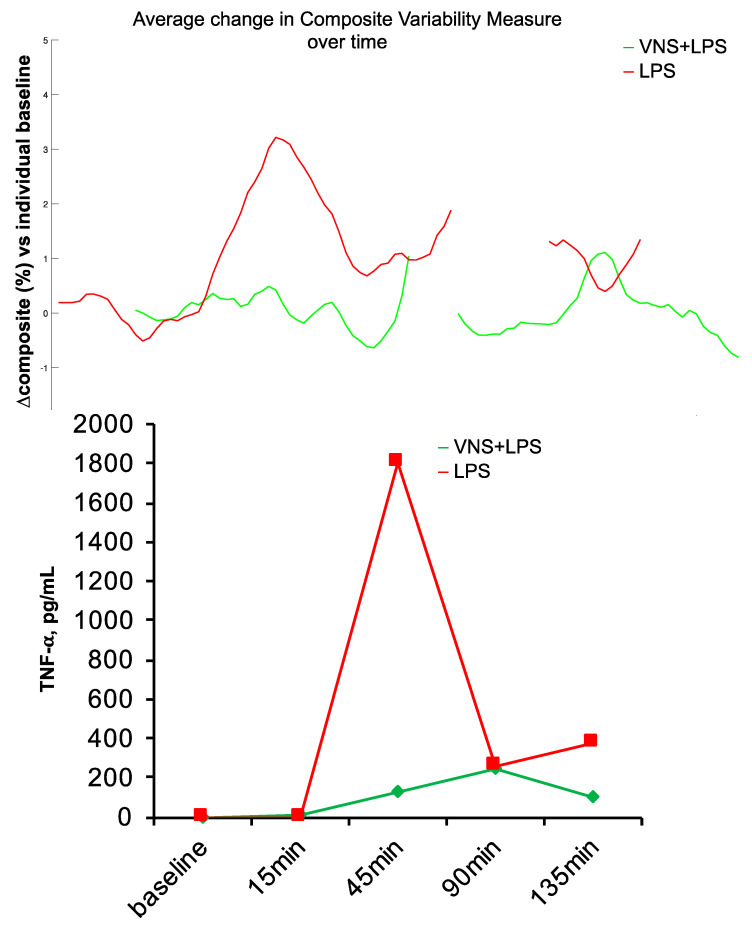
Species universality. Fetal sheep HRV inflammatory response index performs well in the neonatal piglet model of sublethal sepsis. BOTTOM: Piglet’s inflammatory response, as tracked by TNF-α, peaked at ~45 min following LPS injection. TOP: Using the model developed in fetal sheep, HRV inflammatory index was computed on the ECG-derived HRV in a neonatal piglet exposed to LPS only and another with additional vagus nerve stimulation (VNS), which reduced inflammation. Note the concomitant tracking of inflammation with HRV index and TNF-α. The HRV inflammatory index was computed continuously from the available electrocardiogram (ECG) data; some ECG data is missing, which is rendered as gaps in the HRV inflammatory index.

## Data Availability

Accompanying code and data can be found on author’s website (https://FraschLab.org) or directly on FigShare and GitHub as follows: https://figshare.com/authors/Martin_Frasch/5754731; https://github.com/martinfrasch/.

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
