# Peer review of "Heart Rate Variability Code: Does It Exist and Can We Hack It?"

_bioengineering, 2023, doi:10.3390/bioengineering10070822_

Round 1
Reviewer 1 Report
The paper analyzes the HRV code processing specific subsets of HRV measures providing reflecting the specific information about the regulation of physiological responses. The paper is poor of scientific contents. I suggest to resubmit the paper by providing experimental results and related discussion to consider again for another revision process.
Good english
Author Response
I think the reviewer for the comments.
This manuscript is a hypothesis-generating evidence-synthesis article. Published experimental data is discussed and brought together to propose a unified perspective. Novel directions of research are then proposed. I do refer to the readily accessible published body of work that provides the experimental results on which the proposed HRV code paradigm is built.
This review cites 76 studies, most of them experimental. The useful elements are the abstraction of these numerous studies to the concept of the HRV code with its stipulated properties of the temporal coding, phase space structure, target and syndrome specificity and universality (i.e., species independence). This is summarized in Figure 1. Of note, each subsection of the review is divided in two elements: review of the experimental literature supporting a given conjecture (a given HRV code property as stated above) and its significance for the proposed application field. As such, the manuscript discusses many experimental results to propose the HRV code concept. I hope this addresses the comment.
Reviewer 2 Report
This article reviewed HRV on the time structure, dimentinality, specificity and university. The author is a very well-known research on the field of HRV. This review is very interesting and it may play an important role on the future research on HRV. There are some suggestions on writing to enhance the understanding for readers.
[1] There are too many first-person tone description in the manuscript.
[2] The evidences of the four structures are somewhat subjective. A more objective description was expected.
[3] The reason why these four structures were investigated was not clear. Is there any other structure not involved? For specific structures, for example time structure, the major factor is sampling rate. Is there any other factors may be considered? Why or why not?
Author Response
I thank the reviewer for the kind and insightful comments.
[1] I appreciate this point.
This is a single-author manuscript, so speaking from plural "us" seemed even more inappropriate than writing in first person. If the reviewer has specific stylistic suggestions in certain portions of the manuscript, where better wording can be chosen, I will be most grateful for any advice and implement it.
[2] Thank for this comment. I agree that any evidence synthesis follows the best judgement of the synthesizing agent, yours truly in this case. So subjectivity cannot be avoided while I make the best attempt I can to propose a parsimonious model of HRV that is hopefully going to be informative of future studies that will help the field move forward. The evidence upon which I draw does come from different sources, so in this sense, we can defend together the notion that there is some combined body of evidence to support the idea of the HRV code.
If I may ask, what sort of more objective description is the reviewer considering that could be brought to bear here?
[3] That is another great comment! I deduced the properties of the purported HRV code from the generic ideas of what a code should incorporate to be referred to as such. After all, biology is full of code concepts at different scales of organization.
When we define a complex system, a parsimonious, ideally orthogonal, approach is desired, much how we think of dimensions in general. In the present case, some clear overlap exists, intentionally so.
The reason for the chosen properties of the HRV code is that I approach this problem as a physiologist and these are the structures/aspects of the HRV code that I am able to identify as supported by the experimental evidence and as conducive to further research and development of health biomarkers using HRV. I hope that answers this great question, at least in part.
Perhaps, philosophers of science, epistemologists, or other researchers will develop this idea further. That is my hope.
For the specific issue of how the time structure idea is justified or supported by the sampling rate, there exists perhaps the largest body of evidence, some of which I cite in the manuscript. The novelty I see is in calling the phenomenon that the sampling rate changes what HRV is the time structure of HRV, itself being a part of the overarching code properties of HRV. What underlies the sampling rate requirement is of course the nonlinear property of HRV as being an emergent time series produced by multiple weakly coupled non-linear oscillators, as has been discussed before at length, so it seemed to not require additional comment in this manuscript. As such, the existence of the temporal code is the emergent property of these systems. The reason the sampling rate is so important to hone in on is due to the advent of numerous ways to record HRV, especially with wearables, so the practical implications of the sampling rate as a tool that impacts what HRV captures, warrant in my mind this special focus. I discuss this in the manuscript.
Round 2
Reviewer 1 Report
According to my opinion the paper does not introduce useful elements for a review. The authors should add works discussing possible experimental results by providing a deeper analyses about method advantages/disadvantages, perspectives, possible improvements and limits. I suggest to add tables and graphs of comparisons by highlighting differences and union points of works. The paper is not suitable for publication.
English Language Level: Good
Author Response
Thank you for your continued review of the manuscript.
The reviewer stated:
"According to my opinion the paper does not introduce useful elements for a review. The authors should add works discussing possible experimental results by providing a deeper analyses about method advantages/disadvantages, perspectives, possible improvements and limits."
Response: Thank you for the continued review of this manuscript and for sharing your insights.
I respectfully disagree with the assessment that this manuscript contains no experimental results and no useful elements for a review. This manuscript does provide experimental results for each subsection and discusses their significance explicitly in a synthesizing review manner. It is not the intent of the article to reprint the experimental results accessible open-access elsewhere. The intent is to synthesize them in a concise manner toward a novel framework, an HRV code. That is precisely what a review does.
The significance/future steps/perspectives/limitations of the methods (e.g., implications of the temporal coding) are discussed in the respective sections of the manuscript.
The reviewer stated:
"I suggest to add tables and graphs of comparisons by highlighting differences and union points of works."
Response: The review contains a summary figure of the proposed concept of HRV code with reference to the respective underlying experimental data.
Note that I used bolded sentences throughout the manuscript, sparingly and intentionally, to highlight precisely the points this reviewer is requesting: key differences/commonalities if the research and the suggested future work needed.
I revised the manuscript throughout now making more use of this approach.
To further address the suggestion about the tables and graphs, I revised the existing Figure 1 by providing clear references to the respective studies cited and discussed in the manuscript.
I made the changes visible in blue font.
Reviewer 2 Report
This article proposed a novel point of view but is slightly "subjective". The written format is not like the traditional academic paper. Is it possible to re-write the manuscript with traditional academic paper tone?
Author Response
Thank you for your continued review of the manuscript.
The reviewer stated:
"This article proposed a novel point of view but is slightly "subjective". The written format is not like the traditional academic paper. Is it possible to re-write the manuscript with traditional academic paper tone?"
Response:
Thank you the reviewer for highlighting the novelty of the point of view made in the article.
If I may ask: What does this reviewer mean by a traditional academic paper tone?
This manuscript is meant to be an evidence synthesis article, a type of review very much practiced in the scientific literature by many leading journals (e.g., Cell Systems, Phil Transactions journals to name some). Google Search produced 3,500,000 articles in this category.
I hope that this perspective on what constitutes a traditional academic paper is useful to enhance the reviewer's view.
To further address this comment, I revised the manuscript by adding more context to Figure 1 by citing clearly the specific experimental studies underpinning the stated HRV code structure.
Round 3
Reviewer 1 Report
The paper is improved, but about my opinion it should discuss more advances and perspectices, by highlithing differences (possibly in tables) and proposing improvements by authors. .Experimetal results are cited but not properly commented and discussed. The review should address more correlated references or possible correlations (for example about VEG/HRV, ASD/HRV and other inflammatory or physiological status).
Good
Author Response
Reviewer 1: The paper is improved, but about my opinion it should
1. discuss more advances and perspectives, by highlighting differences (possibly in tables) and proposing improvements by authors.
Response:
Thank you.
I revised accordingly so each article subsection contains a subtitled paragraph on the significance and proposed future studies (subsections 2 and 3 in each of the four sections on HRV code properties, respectively).
I added Figures for each of these four subsections to highlight the points.
2. Experimental results are cited but not properly commented and discussed.
Response:
Thank you. I hope this issue has been addressed with the addition of figures for some of the key experimental results.
Specifically,
A) in the new Figure 3, I show the experimental results of the deciphering of HRV inflammatory code;
B) in Figure 4, I walk the reader through the findings of a machine learning model that identified a set of HRV metrics identifying children with various neuropsychiatric disorders, such as ASD and depression, from five minutes of ECG-derived HRV;
C) in Figure 5, I present the findings of HRV inflammatory code developed in fetal sheep model applied to a neonatal piglet model of sepsis.
3. The review should address more correlated references or possible correlations (for example about VEG/HRV, ASD/HRV and other inflammatory or physiological status).
Response:
Thank you.
I added a series of figures for each HRV code property, including data on ASD/HRV study (Fig. 4) and correspondence of HRV inflammatory code (Fig. 3) and between fetal sheep and neonatal piglet (Fig. 5).
I also added references pertaining to relationships of HRV and memory of inflammation or chronic hypoxia as well as a Python code implementation for future studies.
Regarding the correspondence of VENG/HRV, I now refer the interested reader to the relevant freely available preprint (to be published in a Springer Nature Neuromethods book series):
Burns, C. L. Herry, K. J. Jean, Y. Frank, C. Wakefield, M. Cao, A. Desrochers, G. Fecteau, M. Last, A. Seely, C. Faure, M.G. Frasch. The neonatal sepsis is diminished by cervical vagus nerve stimulation and tracked non-invasively by ECG: a preliminary report in the piglet model. arXiv:2002.04136 [q-bio.TO]. Springer Nature Neuromethods Series. In press.
I also refer the interested reader for details on HRV metrics to Herry CL et al. 2019 Physiol. Meas. 40 065004 and Frasch MG Methods X, 2022; 9: 101782. Of note, the latter open-source article also contains a freely accessible Python code for further studies using one of the most comprehensive HRV pipelines available today.